

# Occupation and life satisfaction among individuals with mental illness: the mediation role of self-reported psychophysiological health

Alexandre Granjard[1,2,3,*], Marko Mihailovic[4,5,*], Clara Amato[2,6], Maryam Kazemitabar[7,8], Franco Lucchese[6,9], Christian Jacobsson[1], Nobuhiko Kijima[10,11] and Danilo Garcia[1,2,12,13,14]

[1] Department of Psychology, University of Gothenburg, Gothenburg, Sweden
[2] Blekinge Centre of Competence, Region Blekinge, Karlskrona, Sweden
[3] Promotion of Health and Innovation (PHI) Lab, Network for Well-Being, France
[4] Department of Psychiatry and Behavioral Sciences, Northwestern University, Chicago, IL, USA
[5] Promotion of Health and Innovation (PHI) Lab, Network for Well-Being, USA
[6] Promotion of Health and Innovation (PHI) Lab, Network for Well-Being, Italy
[7] Department of Psychology, University of Tehran, Tehran, Iran
[8] Promotion of Health and Innovation (PHI) Lab, Network for Well-Being, Iran
[9] Department of Dynamic and Clinical Psychology, University of Rome "La Sapienza", Rome, Italy
[10] Faculty of Business and Commerce, Keio University, Tokyo, Japan
[11] Promotion of Health and Innovation (PHI), Network for Well-Being, Japan
[12] Department of Behavioral Sciences and Learning, Linköping University, Linköping, Sweden
[13] Centre for Ethics, Law and Mental Health (CELAM), University of Gothenburg, Gothenburg, Sweden
[14] Promotion of Health and Innovation (PHI) Lab, Network for Well-Being, Sweden
* These authors contributed equally to this work.

Corresponding authors
Alexandre Granjard,
alexandre.granjard.beolet@gmail.com
Danilo Garcia,
danilo.garcia@icloud.com

## ABSTRACT

**Background:** Unemployment can diminish physical, psychological and social health. In this context, research shows that people with mental illness have even more difficulties finding occupation. Thus, some countries, such as Sweden, strive after creating job opportunities for this specific group. We investigated the effect of having an occupation on life satisfaction among individuals with mental illness and whether self-reported physical and psychological health mediated the relationship between being (un)employed and life satisfaction.
**Method:** Two-hundred eighty-seven individuals (148 males, 134 females, and 5 missing information) with mental illness, who received support and services from Swedish Municipalities in Blekinge, self-reported occupation, life satisfaction, and physical and psychological health.
**Results:** Participants who reported having an occupation reported also significantly higher levels of life satisfaction, physical health, and psychological health compared to those without occupation. Nevertheless, these differences were rather small ($Eta^2 < 0.06$). Moreover, the indirect effect of having an occupation on life satisfaction through physical and psychological health was significant. Finally, the total indirect effect of physical and psychological health (i.e., psychophysiological health) accounted for 53% of the total effect of having an occupation on life satisfaction.

**Conclusion:** For individuals with mental illness there seems to be an almost equal importance of indirect and direct effects of having an occupation on their levels of life satisfaction. More specifically, while there are differences in life satisfaction within this population in relation to having an occupation, having an occupation leads to the sense of good psychophysiological health, which in turn helps individuals with mental illness to feel satisfied with their lives.

## INTRODUCTION

Unemployment can lead to, among other things, diminished social status, financial debt, reduced self-esteem, and feelings of guilt. Most importantly, unemployment is significantly associated with psychiatric problems and approximately 37% higher risk of suicide (*Milner, Page & Lamontagne, 2014*). For this reason, there are many interventions aimed to help unemployed individuals to preserve their physical and psychological health during this period (e.g., *Cloninger et al., in press*). Health, however, encompasses also positive outcomes, such as satisfaction with one's life, which is a good proxy for people's well-being (*Diener, Lucas & Oishi, 2018*). In this context, the effect of having an occupation has probably a greater effect on individuals who already have psychiatric problems and who, therefore, have greater difficulty in finding, getting and maintaining a job. Indeed, some countries, such as Sweden, create policies that promote job opportunities for this specific group (https://sweden.se/society/swedens-disability-policy/).

Employment appears to be an important factor in the recovery of individuals with mental illness (e.g., major depression, bipolar disorder, and schizophrenia) (*Dunn, Wewiorski & Rogers, 2008*). Previous research, for example, shows that having an occupation leads to community and social integration, which is a vital part for recovery and rehabilitation; because it enables individuals to live their own life and to perform desired social activities. In other words, by having an occupation a person moves out of their self-stigmatized role and protective environments, towards community and social integration (*Bond et al., 2004*). Nevertheless, society often accepts and assumes a non-working lifestyle for people with mental illness. This might lead to further stigmatization and marginalization, thus, affecting their health and diminishing life satisfaction. Accordingly, only 10–20% of people with mental illness are actually employed in Europe (*Marwaha et al., 2007*). This level of employment among individuals with mental illness appears to be a global trend. In the USA, for example, only 10–15% of this population is employed. Independently of which country is in focus, research shows that being employed or having an occupation is associated with reduced psychological distress and higher quality of life (*Murphy & Athanasou, 1999*). That being said, in Sweden, as much as 30% of people with mental illness are actually employed. Still, 70% are unemployed and more marginalized compared to people in other disability groups. Moreover, only 8% have paid employment and 60% have no occupations at all

(*National Board of Health & Welfare, 1998*). In this context, having an occupation is seen as a critical milestone for physical and psychological recovery, as well as an important factor for life satisfaction.

In short, inactivity and loss of productive years are associated to mental illness. What is more, individuals with mental illness become discouraged when they fail finding or maintaining a job. Indeed, unemployment alone can lead to deterioration in physical and psychological health even among previously healthy individuals. This effect is more accentuated for those individuals who had mental illness to begin with (*Corrigan & McCracken, 2005*) and who often present higher prevalence of health-related risks (*Blomqvist et al., 2018a*; *Lundström et al., 2017*). Having an occupation is expected to help to reintegrate individuals with mental illness and disabilities into their communities, because having an occupation decreases self-stigma and increases the sense of being physically and psychologically healthy. For these reasons, improving vocational rehabilitation services has become a priority for the promotion of quality of life, life satisfaction, and health per se (*Bond et al., 2004*). Nevertheless, the question is if having an occupation is related to the physical and psychological health of individuals with mental illness or if having an occupation per se increases life satisfaction? Many policies in this regard are actually based on the assumption that having an occupation directly leads to both health and life satisfaction, despite the fact that the association between health and life satisfaction is not straightforward (*Rohrer & Lucas, 2020*). For instance, there are many confounding factors that affect both health and life satisfaction (*Rohrer & Lucas, 2020*). To the best of our knowledge, however, no studies have addressed these relationships within a Swedish population with mental illness.

## The present study

Our aim was to investigate the effect of having an occupation on self-reported life satisfaction among Swedish individuals diagnosed with mental illness who receive support and care from the Municipalities. We also investigated whether self-reported physical and psychological health mediated the relationship between being employed and life satisfaction within this population. Our specific hypotheses were:

Hypothesis 1: Individuals with mental illness with an occupation are more satisfied with their lives compared to those who do not have an occupation.

Hypothesis 2: Individuals with mental illness with an occupation perceive themselves as healthier, both physically and psychologically, compared to those who do not have an occupation.

Hypothesis 3: Physical and psychological health mediate the relationship between having an occupation and life satisfaction among individuals with mental illness.

## METHOD

### Ethical statement

The study was performed in accordance with the ethical standards of the 1964 Helsinki declaration and its later amendments. Participants were provided with the aims of the study, that participation was anonymous and voluntary, that they had the opportunity to

ask questions, and that they were free to withdraw at any time without giving a reason and without cost or any repercussions regarding the services, care or support they received from the Municipalities. Verbal consent was obtained and the study was approved by the Swedish Ethical Review Authority (Dnr. 2017/305).

## Participants and procedure

The present study was conducted in the fall of 2017 at the Blekinge Centre of Competence, the research and development unit of Region Blekinge, Sweden. The five Blekinge Municipalities were involved in the study: Sölvesborg, Olofström, Karlshamn, Ronneby, and Karlskrona. All 621 individuals between 18 and 65 years of age who received support[1] due to mental illness in each of the five municipalities in Blekinge were contacted. The staff working closest to the clients were the ones responsible for the exclusion procedure. A total of 146 individuals were excluded from the study because they suffered from dementia or were using drugs at the time the survey was distributed. Out of those 475 eligible to participate, 62 chose to not participate in the study, 126 did not respond to all survey questions, and four of them didn't indicate if they had a job or not, and were therefore excluded from the analyses. That is, the final sample represented roughly 60% of those eligible to participate: 287 individuals (148 males, 134 females, and 5 missing) with a mean age of 43.46 years (SD = 13.32). About 3.5% of the participants did not finish primary school, 23.7% finished primary school, 52.3% had a high-school degree, 12.2% had higher education and 8.3% had other type of education. Most of the participants were single (76.3%) and lived in their own accommodation (74.6%) and only a few of them reported living in an institution for individuals with mental illness (16.4%). A total of 112 of the participants reported not having any occupation, whereas 171 reported having an occupation.

## Measures

### Life satisfaction

We used the Swedish version (*Garcia & Siddiqui, 2009*) of the Satisfaction with Life Scale (*Diener et al., 1985*), which comprises 5 items (example item is: "In most ways my life is close to my ideal") that are rated on a 7-point Likert scale (1 = *strongly disagree*, 7 = *strongly agree*). The life satisfaction score was computed as the sum score of the 5 items divided by 5. Consistent with previous findings with the original English version (e.g., *Diener, Inglehart & Tay, 2013*), the Swedish version showed good internal consistency in the present study (α = 0.86).

### Occupation

Participants were asked to indicate if they had an occupation or not (yes/no). A total of 112 of the participants reported not having any occupation, whereas 171 reported having an occupation.

### Physical and psychological health

Health was assessed with a 4-point Likert scale (1 = *bad*, 4 = *excellent*) and two single items: "How would you rate your physical health in general?" and "How would you rate your

[1] The support includes: help with everyday finances, help with daily shores, transport, support when contacting authorities, help taking social contact, support seeking job or occupation, and help with lifestyle habits (https://sweden.se/society/swedens-disability-policy/).

mental health, in general?". The overall self-reported physical health of the sample was $M = 2.30$ (SD = 0.84) and the overall self-reported psychological health was $M = 2.26$ (SD = 0.80).

## Statistical treatment

We only used the responses of those individuals who answered to all variables included in the analyses ($n = 260$). In order to test Hypotheses 1 and 2, an Analysis of Variance (ANOVA) was performed using occupation as the independent factor and self-reported life satisfaction, physical health, and psychological health, as the dependent variables. In order to test hypothesis 3, we used *Preacher & Hayes (2008)* procedure to extrapolate estimates of direct and indirect effects. In presence of a multiple mediation model, this procedure is particularly recommended because it allows researchers to determine not only whether an overall indirect effect exists, but also to compare the specific indirect effect of one mediator in the presence of other mediators in the model (*Preacher & Hayes, 2008*). As recommended, we ran the multiple mediation model using bootstrapping (i.e., a nonparametric resampling procedure; in our case, 5.000 resamples) with 95% confidence intervals (CIs). All analyses were conducted in IBM SPSS version 24.

# RESULTS

## Differences in self-reported life satisfaction and psychophysiological health

Regarding self-reported life satisfaction, the analysis of variance revealed a significant difference in self-reported life satisfaction ($F = 8.162$, $p < 0.01$; $Eta^2 = 0.031$) between individuals who reported having an occupation ($M = 3.60$) compared to those without occupation ($M = 3.07$). In addition, self-reported physical health was significantly ($F = 7.481$, $p < 0.01$; $Eta^2 = 0.028$) higher among individuals with an occupation ($M = 2.42$) compared to those without occupation ($M = 2.14$). As expected, self-reported psychological health was significantly ($F = 5.353$, $p < 0.05$; $Eta^2 = 0.020$) higher among individuals with an occupation ($M = 2.34$) compared to those without one ($M = 2.11$). Nevertheless, these differences were rather small—all with an $Eta^2 < 0.06$.

## The mediational effect of self-reported psychophysiological health on the relationship between occupation status and self-reported life satisfaction

The mediational model was significant ($F_{(3, 256)} = 35.90$, $p < 0.001$, $R^2 = 0.54$, Adj. $R^2 = 0.30$). More specifically, the results showed that both mediators, self-reported physical ($\beta = 0.40$; $p < 0.001$) and psychological health ($\beta = 0.70$; $p < 0.001$), were significantly positively associated to life satisfaction. In addition, the total effect of occupation on self-reported life satisfaction was positive and significant (total effect = 0.53 with BCa 95% CI of [0.16–0.88]) and the total indirect effect of having an occupation on self-reported life satisfaction was also significant (indirect effect = 0.28, with BCa 95% CI of [0.09–0.49]). Moreover, the indirect effects of having an occupation on life satisfaction both through physical health (indirect effect = 0.11, with BCa 95% CI of [0.03–0.23]) and psychological

**Table 1 Partially standardized indirect effects of having an occupation on life satisfaction through self-reported physical and psychological health.**

| | Bootstrap effect | SE | Bias corrected and accelerated 95% CI | |
|---|---|---|---|---|
| | | | Lower | Upper |
| Total effect | 0.19 | 0.07 | 0.06 | 0.33 |
| Physical health | 0.08 | 0.03 | 0.02 | 0.15 |
| Psychological health | 0.11 | 0.05 | 0.02 | 0.21 |
| Contrast for indirect effects | | | | |
| Psychophysiological health | −0.03 | −0.02 | 0.00 | −0.06 |

Note:
 CI, Confidence Intervals; Psychophysiological Health stands for the effect of both self-reported Physical and Psychological Health.

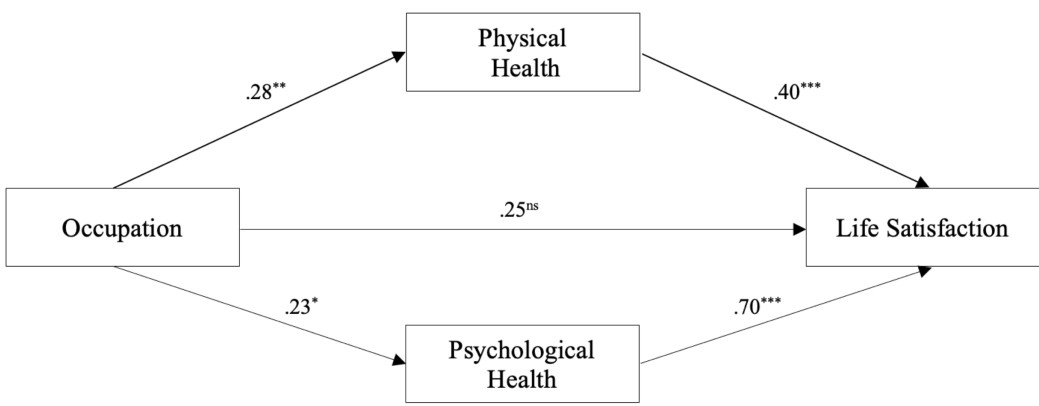

**Figure 1 Coefficients representing the effects of occupation on the mediators (i.e., self-reported physical and psychological health) and self-reported life satisfaction.** $^{*}p < 0.05$; $^{**}p < 0.01$; $^{***}p < 0.001$; ns = nonsignificant.

health (indirect effect = 0.16, with BCa 95% CI of [0.03–0.32]) were significant. The contrast among the two indirect effects was non-significant, thus, indicating that the magnitude of the mediators' effects on self-reported life satisfaction was equal. In other words, the two mediators contributed equally to the effect of having an occupation on self-reported life satisfaction. Finally, the total indirect effect of physical and psychological health (i.e., psychophysiological health) accounted for 53% of the total effect of having an occupation on life satisfaction, thus, suggesting an almost equal importance of indirect and direct effects. See Table 1 and Fig. 1 for the details.

## DISCUSSION

We investigated how having an occupation is related to self-reported physical and psychological health and, in turn, to life satisfaction among individuals with mental illness who receive care and support from Municipalities in Blekinge, Sweden. Our results showed that individuals who have an occupation (vs. those who do not have one) reported being more satisfied with their lives, and perceived themselves as being in a better physical condition (i.e., physical health) and having a better mental health (i.e., psychological

health). In turn, this positive perception of themselves increased a more general feeling of satisfaction with life.

Individuals with mental illness are more sensitive to the negative effects of unemployment and experience stronger difficulties to find work, mostly due to existing stigma from society (*Bond et al., 2004*), but also from themselves (*Dunn, Wewiorski & Rogers, 2008*). Once they are able to secure employment, their self-perception improves, which reflects positively on the apprehension of their own physical and psychological health. On the other hand, the opposite relation can also be expected—individuals with mental illness who perceive themselves having good physical and psychological health, have better chances to find employment and making self-directed choices with regard to a healthy life style (*Lundström et al., 2019*). Indeed, our cross-sectional design does not allow us to scrutinize causal inference (see *Rohrer & Lucas, 2020*). Moreover, the differences within this population, between individuals with an occupation and those without one, had rather small effect sizes. Nevertheless, employment definitely helps individuals with mental illness, who struggle to have a steady stream of income, to secure their basic human needs, such as, housing (*Bond et al., 2004*; *Henwood et al., 2015*). Indeed, being employed and having an income might provide this population with an opportunity to access better quality of life, more stable housing, and also give them a sense of personal, emotional and financial security and safety, which in turn might influence overall well-being (*Dunn, Wewiorski & Rogers, 2008*; *Henwood et al., 2015*). Furthermore, employment supports individuals with mental illness to meet needs such as social connectedness and self-esteem, which particularly improves their perception of their own psychological health (*Dunn, Wewiorski & Rogers, 2008*).

In this context, having an occupation and being part of a group promotes the sense of social acceptance and belongingness (*Dunn, Wewiorski & Rogers, 2008*; *Bond et al., 2004*). This might help individuals with mental illness to meet the essential human need of being part of a larger community (*Henwood et al., 2015*). This population struggle with social inclusion and acceptance due to a widely present prejudice and discrimination towards individuals with mental illness (*Bond et al., 2004*). These negative social factors contribute to developing a sense of isolation and loneliness that ultimately lead to negative emotions such as anxiety and depression followed by deterioration of overall mental and psychological health (*Murphy & Athanasou, 1999*). In contrast, the sense of social belongingness and community connectedness instills hope and love, which positively influences their perception of psychological health and ultimately, as shown here, leading to overall life satisfaction (*Dunn, Wewiorski & Rogers, 2008*; *Bond et al., 2004*).

Nevertheless, individuals with mental illness have difficulties with self-acceptance, due to the nature of their illness (*Dunn, Wewiorski & Rogers, 2008*) and the social hindrances they face, such as, stigma, prejudice, and discrimination (*Bond et al., 2004*). Indeed, most humans benefit from feeling respected, accepted, and valued by and being in the service of others (*Roberts et al., 2007*; *Cloninger, 2004*). In short, as other studies show, employment might provide this population with a sense of contribution and value and even self-acceptance (*Dunn, Wewiorski & Rogers, 2008*) and also a sense of personal achievement including respect from others, recognition, attention, self-efficacy and

self-esteem (*Dunn, Wewiorski & Rogers, 2008*; *Bond et al., 2004*). Thus, individuals with mental illness who are employed are likely to increase their own trust on their capabilities (self-efficacy), to feel valued (self-esteem), and to have a positive outlook about their personal as well as professional future. Here however, we showed that having an occupation might not only improve their material life (e.g., income), their social life, and their trust in their own capacity; but also that it is positively related to their own perception about their physical and psychological health, which in turn is positively related to their life satisfaction.

That all being said, besides the limitations detailed above, our study had others caveats. Mainly, the question of selection bias. Since the selection of participants was done by the staff working closest to them, proper randomization was definitely not achieved. In other words, even if we had a modest participation in our study (i.e., about 45.6% of all 621 individuals between 18 and 65 years of age who received psychiatric support by the public services in Blekinge), those who participated might not be representative for the population of individuals with mental illness. Moreover, we did not include gender, age or education as covariates in the relationship between having an occupation and self-reported life satisfaction and psychophysiological health. Thus, future studies need to take this into consideration. After all, the relationship between health and the different components of subjective well-being (e.g., life satisfaction) is complex (*Rohrer & Lucas, 2020*).

## CONCLUSIONS

Individuals with mental illness who are employed, compared to those who are unemployed, reported slightly higher levels of overall life satisfaction and psychophysiological health. More importantly, the mechanism underlying the relationship between having an occupation and life satisfaction implies that having an occupation enhances individuals' perception of their own physical and psychological health, which in turn might increase their satisfaction with life. The causal effect, however, is still not clear. It is possible that individuals who perceive themselves as having better health are the ones who are able to find and keep a job, which in turn, leads to life satisfaction. Moreover, there are many possible underlying causes to both psychological and physical health and life satisfaction that are difficult to control for (cf. *Rohrer & Lucas, 2020*). Thus, policies based on the notion that having an occupation is directly associated to people's health and life satisfaction might be wrong. In other words, we might need to create activates that target the root of mental illness rather than the symptoms (e.g., unemployment).

Importantly, previous research shows that purpose and meaning in life in the general population moderates the relationship between income satisfaction and life satisfaction (*Joshanloo, 2018*). Individuals who believe their lives have meaning and purpose are likely to deemphasize the role of materialistic aspirations, focus more on intrinsic pursuits, and have more mental resources to cope with financial challenges (*Joshanloo, 2018*). These individuals focus on positive outcomes such as satisfaction with one's life as a whole rather than focusing on particular external factors such as employment. Interviews among individuals with mental illness, for instance, mirror a wish of being accepted by the self and others (i.e., "Being regarded as a whole human being by self and others") and the need

of everyday structure, motivating life events, and support from significant others (*Blomqvist et al., 2018b*). In this context, person-centered programs including biopsychosocial services (e.g., well-being coaching), that focus on improving health-related capacities, such as, self-directedness (e.g., self-acceptance, goal-directedness, resourcefulness), cooperativeness (e.g., empathy, helpfulness, social tolerance), and self-transcendence (e.g., flow, meaning in life, spiritual acceptance, unity); are beneficial for different long-term unemployed populations (*Cloninger et al., in press*). Such biopsychosocial services account for the multidimensional nature of health and provide the evidence-based tools that are needed to alleviate the suffering related to individual's mental illness, and to cope with both unemployment and other challenges of the 21st century.

## ACKNOWLEDGEMENTS

We would like to thank the participants and the staff at the five Municipalities in Blekinge for making this research possible. We would also like to thank Ledningssamverkan vård och omsorg (LSVO) in Blekinge for allowing this research. Last but not the least, we want to express our gratitude to Lars-Henry Gustle, Ulrika Harris, and Carina Ström from FoU-avtalet, who designed and conducted the data collection together with other colleagues at the Centre of Competence, Region Blekinge. Ulrika Harris also helped us by reading and commenting a first version of the article.

### Funding

The project was supported by Region Blekinge and the five Municipalities of Blekinge (Sölvesborg, Olofström, Karlshamn, Ronneby, and Karlskrona) through their Research and Development agreement (i.e., FoU-avtalet). The funders had no role in study design, data collection and analysis, decision to publish, or preparation of the manuscript.

### Grant Disclosures

The following grant information was disclosed by the authors:
Region Blekinge and the five Municipalities of Blekinge (Sölvesborg, Olofström, Karlshamn, Ronneby, and Karlskrona).

### Competing Interests

The authors declare that they have no competing interests.

### Author Contributions

- Alexandre Granjard performed the experiments, analyzed the data, prepared figures and/or tables, authored or reviewed drafts of the paper, and approved the final draft.
- Marko Mihailovic performed the experiments, authored or reviewed drafts of the paper, and approved the final draft.
- Clara Amato conceived and designed the experiments, performed the experiments, analyzed the data, prepared figures and/or tables, authored or reviewed drafts of the paper, and approved the final draft.

- Maryam Kazemitabar performed the experiments, analyzed the data, prepared figures and/or tables, authored or reviewed drafts of the paper, and approved the final draft.
- Franco Lucchese performed the experiments, authored or reviewed drafts of the paper, and approved the final draft.
- Christian Jacobsson performed the experiments, authored or reviewed drafts of the paper, and approved the final draft.
- Nobuhiko Kijima performed the experiments, authored or reviewed drafts of the paper, and approved the final draft.
- Danilo Garcia conceived and designed the experiments, performed the experiments, analyzed the data, prepared figures and/or tables, authored or reviewed drafts of the paper, and approved the final draft.

### Human Ethics

The following information was supplied relating to ethical approvals (i.e., approving body and any reference numbers):

The study was approved by the Swedish Ethical Review Authority (Dnr. 2017/305).

### Data Availability

Raw measurements are available as a Supplemental File.

### Supplemental Information

Supplemental information for this article can be found online at http://dx.doi.org/10.7717/peerj.10829#supplemental-information.

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
