# Peer review of "Occupation and life satisfaction among individuals with mental illness: the mediation role of self-reported psychophysiological health"

_PeerJ, doi:10.7717/peerj.10829_

## Round 0.1 · original submission · Minor Revisions

Dear Authors:

The reviewers and I all feel this is an important area of study and a potential contribution to the field, but that there are revisions needed to make it as clear and as impactful as possible.

Please revise and make a careful note of the revisions in your rebuttal letter so that the changes are easy to follow. In addition, some of the language is awkward, Reviewer 2 made some concrete suggestions but I advise having the complete manuscript once you have made the suggested revisions edited by a native English speaker.

Reviewer 1 ·

Basic reporting

It comes as a surprise, that there are a lot of authors involved in this paper, especially since not all of them are affiliated in Sweden (i.e. Japan, Iran). It might be useful to quote the “authors´ contribution”, since this is a standard in international journals.

Experimental design

It would be interesting to get information about the kind of psychiatric support people get and what kind of occupation they had, since this might have an impact on the results (line 132ff).

It is not clearly described, based on which criteria the socials workers have made the decision, who to excludes from the sample (line 135f).

Validity of the findings

Raw data were supplied, but the central item concerning occupation is missing, so further examinations and proofs of the findings could not be done.

Declare, which statistical program has been used, such as SPSS, SAS or other programs (LISREL). In SPSS there is a tool for mediation analysis, that may be used (Field, A. 2013, Discovering Statistics using IBM SPSS Statistics, 4th Edition, p408ff).

Please describe how the variable “qol” (quality of life) is computed i.e. weighted or sum scores, as this is not comprehensible yet.

It might have been useful to include covariates (gender, age, education) in the analysis, as these control variables proof correlation effects of the mediation variable. If this analysis was done, it should be commented on.

Additional comments

The paper describes an important and relevant topic. It uses throughout a clear and professional English language and the structure conforms to international standards.
The paper is easy to follow: Research design is clearly structured, well defined, relevant and meaningful. Tables and graphs support the key messages. Key results are well summarised and the authors point out the limitations of the study. Literature is up to date and research is within the scope of PeerJ.

Reviewer 2 ·

Basic reporting

Firstly, the language used in the manuscript is mostly unambiguous, professional and helps to understand the study. However, some uses of the language should be adapted to the general meaning because it creates some strange structures (for example, line 83: diminishing not only their health (maybe affecting their health and diminishing life satisfaction levels; line 166: physical health was significantly (maybe using physical and psychological health’s perception because they came from self-reported measures) and also avoid the use of contractions throughout the manuscript (line 138).

Secondly, although the references used are adequate and appropriate, more references from the last 5 years could be adequate, if possible, to create a more updated theoretical background. Additionally, more information about previous pieces of research or the absence of these studies analyzing the relationship between the variables of the study will be needed in the introduction in order to clarify the need of investigating this topic. In this sense, it would be appropriate to justify the hypotheses with references. Besides, some references used throughout the manuscript are missing at the references section (Cloninger, 2004) ore the year is not the same (Henwood et al., 2014).

Regarding the raw data included, it contains the appropriate information. However, I have identified that in the raw data document there are 287 individuals and in the manuscript you have mentioned that 283 participated. Probably, there is a misunderstanding here. Try to clarify it.

Figures and tables used are adequate. In Figure 1 it could be better to change the concept job by occupation used in the manuscript.

Finally, there are some headings with full stops (line 129).

Experimental design

A Data analyses section will be needed before the Results section in order to clearly specify the analyses performed and the statistical analyses programs used.

In the Discussion section, the importance of filling the knowledge gap that must be identified in the introduction should be included here and the results can be discussed using the hypotheses mentioned in the introduction.

Validity of the findings

No comment.

Additional comments

I really appreciate the effort of analyzing an interesting and important topic. I hope that with the suggestion indicated the manuscript quality will be improved.

---

## Round 0.2 · accepted · Accept

Well done - your paper will contribute to our understanding of psychiatric rehabilitation and recovery